# Aggregating multiple test results to improve medical decision-making

**Lucas Böttcher** [1] *, **Maria R. D'Orsogna** [2,3], **Tom Chou** [3,4]

**1** Department of Computational Science and Philosophy, Frankfurt School of Finance and Management, Frankfurt am Main, Germany, **2** Department of Mathematics, California State University at Northridge, Los Angeles, California, United States of America, **3** Department of Computational Medicine, University of California, Los Angeles, Los Angeles, California, United States of America, **4** Department of Mathematics, University of California, Los Angeles, Los Angeles, California, United States of America

* l.boettcher@fs.de

**Data Availability Statement:** Our source codes are available at https://gitlab.com/ComputationalScience/testing-statistics.

**Funding:** We acknowledge financial support from hessian.AI (LB), the ARO through grant W911NF-

## Abstract

Gathering observational data for medical decision-making often involves uncertainties arising from both type I (false positive) and type II (false negative) errors. In this work, we develop a statistical model to study how medical decision-making can be improved by aggregating results from repeated diagnostic and screening tests. Our approach is relevant to not only clinical settings such as medical imaging, but also to public health, as highlighted by the need for rapid, cost-effective testing methods during the SARS-CoV-2 pandemic. Our model enables the development of testing protocols with an arbitrary number of tests, which can be customized to meet requirements for type I and type II errors. This allows us to adjust sensitivity and specificity according to application-specific needs. Additionally, we derive generalized Rogan–Gladen estimates of disease prevalence that account for an arbitrary number of tests with potentially different type I and type II errors. We also provide the corresponding uncertainty quantification.

## Author summary

Our work focuses on medical decision-making, particularly on addressing uncertainties associated with screening and diagnostic tests. No test is perfect, so finding a balance between false positives (misidentifying a condition) and false negatives (missing a condition) is crucial in many biomedical applications. Implementing accurate and efficient testing is important not only for individual diagnoses but also for population-wide testing during a pandemic. Since cost-effective and rapid tests are often quite inaccurate, a common goal is to obtain accurate assessments from repeated testing and meaningfully combining their results. However, using the multitude of tests and their different sequences of administration to design effective test protocols is a challenge that requires new statistical tools. In this study, we develop tools for aggregating test results in ways that can be tailored to specific applications by tuning the false positive-false negative ratio. Furthermore, we demonstrate how our method can improve

23-1-0129 (MRD and LB), and the NSF through grant OAC-2320846 (MRD). The funders had no role in study design, data collection and analysis, decision to publish, or preparation of the manuscript.

**Competing interests:** The authors have declared that no competing interests exist.

disease prevalence estimates and thus aid in the implementation of effective public-health measures.

## Introduction

Administering effective diagnostic and screening tests plays an important role in most biomedical decision-making. Recent advancements in biotechnology have made a wide array of biochemical tests readily available on a large scale. For example, in the case of SARS-CoV-2, a systematic review identified 49 different antigen tests [1] which are cost-effective and can provide results in 15–30 minutes. However, their sensitivity (*i.e.*, true positive rate) can be as low as 34.3% in symptomatic patients and 28.6% in asymptomatic patients [1]. This indicates that some tests correctly identify an infected individual as positive in only about one third of cases, leaving a significant portion of those with the disease undetected. Besides sensitivity, another metric used to assess the accuracy of a test is its specificity (*i.e.*, true negative rate). Highly sensitive tests prioritize identifying individuals with a disease, while highly specific tests prioritize identifying those who do not have the disease. In most cases, sensitivity and specificity are inversely related; both are important when assessing the value of a medical test [2, 3].

Given the availability of various tests with differing sensitivities and specificities, how can one repeat tests and integrate results to minimize both type I errors (false positives) and type II errors (false negatives)? Although this is a key question across many different clinical settings, including diabetes testing [4, 5], medical imaging [6–8], prostate cancer testing [9], and stool sample analysis in colon cancer testing [10, 11], our primary focus will be on aggregating results from different tests within the context of SARS-CoV-2 due to the availability of comprehensive studies on properties of the corresponding tests.

The SARS-CoV-2 pandemic emphasized the crucial role of testing in managing the spread of an infectious disease. During the early stages of the pandemic, shortages of test kits were common, causing delays in diagnoses and leading to underreporting of COVID-19 cases which hindered the effectiveness of public-health measures. Due to the intensifying crisis, regulatory agencies expedited the review and approval process of dedicated tests developed by different suppliers in different countries that used different technologies. These different tests were distributed and used at the same time.

One distinguishes between two primary categories of SARS-CoV-2 tests: (i) viral tests and (ii) antibody (or serological) tests [12]. Within the viral-test category, there exist two main subclasses: nucleic acid amplification tests (NAATs), such as reverse transcription polymerase chain reaction (RT-PCR) tests that typically detect viral RNA, and antigen tests that detect specific antigen proteins on the surface of the virus. Antibody tests serve to identify antibodies produced as part of the adaptive immune system response. In the context of SARS-CoV-2, antibody tests may target anti-nucleocapsid antibodies, indicative of current or past infection, and anti-spike protein antibodies, generated through infection or vaccination.

Early detection methods relied on genetic sequencing and RT-PCR tests to detect viral genetic material. Antibody tests were also introduced to detect the presence of the virus in previously infected individuals who had developed an immune response. Both tests required specialized laboratory equipment and personnel to process them, making the diagnosis of active infections (RT-PCR tests) or of an activated immune response (antibody tests) available only after a few hours or even days. As the pandemic surged, the prioritization of rapid testing methods led to the development of rapid antigen tests, capable of detecting viral proteins and providing results within minutes. Subsequent saliva-based tests offered a less invasive

experience compared to those based on nasopharyngeal swabs. Finally, the retreat of the pandemic was accompanied by the introduction of home testing kits. Current research is focused on perfecting new methods, including breathalyzer tests and wastewater monitoring.

Each testing method has its specific advantages and limitations. For example, RT-PCR tests are highly sensitive and specific and can detect even small amounts of viral RNA. However, there may be long delays in obtaining actionable results. Antibody tests may not detect antibodies in the early stages of the infection and are prone to large false-positive results due to cross-reactivity with antibodies from other viruses. Antigen tests are usually less sensitive than RT-PCR tests, but provide results quickest. Further variability in sensitivity and specificity arises within each type of testing method due to differences among test manufacturers and periodic modifications to the biochemical protocols, which are made to ensure the detection of any novel viral mutations or variants.

Our collective past experience with the spread of the SARS-CoV-2 virus poses several challenges in preparing and responding to future pandemics, including how to best allocate scarce resources and enhance testing and classification strategies. The development of appropriate mathematical and computational methods plays a fundamental role in addressing these challenges. For example, one way to stretch resources is to test pooled samples, allowing one to eliminate large numbers of uninfected individuals with a small number of tests. Several mathematical approaches have been developed to study the optimization of both sample pooling and testing [13]. These approaches consider factors such as test sensitivity and specificity [14], estimated prevalence [15–17], disease dynamics [18], and available social contact information [19]. Other mathematical techniques aimed at improving testing efficiency by accounting for uncertainty in disease prevalence [20], indeterminate test results [21], time-dependent prevalence and antibody levels [22, 23], high-dimensional data analysis to improve classification accuracy [24], and multiple classes such as vaccinated, previously infected, and unexposed individuals [25].

In this paper, we focus on developing mathematical and computational methods that can help improve medical decision-making by repeating tests and aggregating their results. We use the term "aggregate" to specifically refer to the process of using Boolean functions to map multiple binary test results to a single binary output. Several related studies have highlighted the potential of this approach [26–35], often using different terms such as "all heuristic" [31, 32], "believe-the-negative rule" [36], "conjunctive positivity criterion" [28, 37, 38], and "orthogonal testing" [39] to refer to the same protocol in which all tests must return a positive result in order to classify an individual as infected. In Boolean algebra [40], this corresponds to an aggregation using the binary AND operator. Another aggregation method is the "any heuristic" [31, 32] also termed the "believe-the-positive rule" [36] or "disjunctive positivity criterion" [28, 37, 38]. In this protocol, all tests must return a negative result in order to classify an individual as not infected. It is thus sufficient for one test to be positive for a positive diagnosis. In Boolean algebra, this aggregation method is represented by the binary OR operator.

The US Food and Drug Administration (FDA) has also recognized the relevance of repeated testing and released an Excel-based calculator to compute properties of two combined tests [41]. However, most available aggregation methods, including the FDA calculator, only consider two tests and usually employ very few (between one and three) Boolean functions. Nevertheless, there are instances where jurisdictions have implemented testing protocols involving three and four tests, such as in Vienna, Austria [42], and Santiago, Chile [43]. Without appropriate mathematical insight and computational tools, however, it is challenging to analyze the properties of all possible aggregation methods due to the vast number of tests and their combinations. The lack of theoretical understanding often results in the implementation of ad-hoc and suboptimal aggregation protocols, rather than the most efficient ones. In

addition to determining the disease status of an individual, combined tests can improve estimates of disease prevalence [27, 44], which is helpful in infectious-disease surveillance and management [38, 45–54]. In this context, it is also important to develop suitable mathematical tools to compare disease-prevalence estimates across jurisdictions, as different public-health organizations employ different testing protocols and aggregation methods [42, 43, 55–57].

Here, we combine concepts from biostatistics and Boolean algebra to develop a broadly applicable statistical model that can guide medical decision-making after repeated screening or diagnostic testing. We show how our model enables the development of testing protocols whose overall sensitivity and specificity can be tuned to satisfy application-specific requirements on type I and type II errors. Additionally, we present an algorithm capable of determining the best way to aggregate results from a given set of tests in terms of efficient sensitivity-specificity pairs. Furthermore, we integrate our aggregation approach with population-level prevalence estimation, demonstrating how repeated testing can enhance prevalence monitoring. Specifically, we generalize the Rogan–Gladen prevalence estimate [27, 44] to account for an arbitrary number of tests, each having potentially different type I and type II error rates.

## Results

### Aggregating results from two tests

As a starting point, we examine testing protocols that combine the results of $n = 2$ tests (possibly of different types), denoted by binary random variables $Y_1$ and $Y_2$, where $Y_1, Y_2 \in \{0, 1\}$. Here, $Y_1, Y_2 = 0$ indicates a negative test result, while $Y_1, Y_2 = 1$ represents a positive test result. The true disease status of an individual, classified as either negative (−) or positive (+), is represented by another binary random variable $X \in \{0, 1\}$.

The true positive rates TPRs (or sensitivities) of each of the two (type 1 and type 2) tests are defined as

$$\text{TPR}_1 = \Pr(Y_1 = 1 \mid X = 1) \quad \text{and} \quad \text{TPR}_2 = \Pr(Y_2 = 1 \mid X = 1), \tag{1}$$

respectively. The corresponding true negative rates (TNRs, or specificities) are

$$\text{TNR}_1 = \Pr(Y_1 = 0 \mid X = 0) \quad \text{and} \quad \text{TNR}_2 = \Pr(Y_2 = 0 \mid X = 0), \tag{2}$$

respectively. We use $\Pr(Y \mid X)$ to denote the conditional probability of $Y$ given $X$.

The individual-test TPRs and TNRs in Eqs (1) and (2) serve as building blocks for modeling the overall TPR and TNR of a testing and aggregation protocol involving multiple tests.

For $n = 2$ ordered tests, there are $r = 2^n = 2^2 = 4$ possible sequences of test results (the permutations of "+" and "−" of length 2): (+, +), (+, −), (−, +), and (−, −). These four sequences can be used as logic inputs to a Boolean function that maps each of them to either a positive (+) or negative (−) assigned disease status. Therefore, there are $2^r = 2^4 = 16$ possible mappings, which corresponds to the total number of two-input Boolean gates. Notice that not all gates are relevant in the context of medical decision-making. For instance, gates that return only positive or negative results would not be practical. Assuming all individual tests have "discriminatory power" (*i.e.*, perform better than a random classifier), one can show that the set of efficient tests is formed by AND, OR, and one of the single tests [38]. A test is considered efficient if no other test performs better in one aspect (sensitivity or specificity) without performing worse in the other.

In Fig 1, we show AND and OR gates for $n = 2$ tests. Depending on how the individual test results are processed, the output of the chosen aggregation function assigns a positive or negative disease status. Both AND and OR aggregation functions have been used in SARS-CoV-2 seroprevalence studies (see Table 1) and we will analyze both in this paper. Notice that the

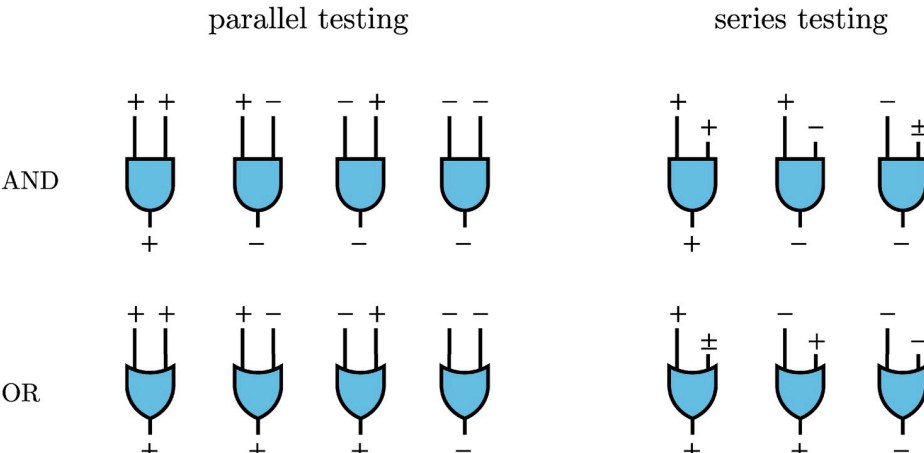

**Fig 1. Parallel and series testing protocols using two tests.** Positive (+) and negative (−) test outcomes are combined using the two Boolean functions AND (⊓) and OR (⊔). In parallel testing, both inputs are assessed simultaneously, while in series testing, the left input is examined before the right. Hence, if the initial test in a series protocol yields a negative result with aggregation through an AND gate, the assigned disease status will be negative, irrespective of the second input. In series testing with an OR gate, the assigned disease status will be positive if the first test is positive, regardless of the outcome of the second test.

aggregated output sequences of the parallel and series testing protocols shown are the same. The only implicit difference lies in how the input signals are processed (parallel or series).

Aggregating test results using an AND gate produces a positive result if and only if both inputs are positive, corresponding to a "conjunctive positivity criterion" [28, 37, 38]. Otherwise the output result is negative. For an OR gate, the aggregate test result is negative if and only if both inputs are negative, corresponding to a "disjunctive positivity criterion" [28, 37, 38]. In the remaining cases, the OR aggregation method outputs a positive result. The process of aggregation is sometimes referred to as the "all heuristic" [31, 32] and the "believe-the-negative rule" [36] when using AND aggregation. Similarly, OR aggregation is sometimes termed the "any heuristic" [31, 32] and the "believe-the-positive rule" [36]. For the two possible test-administration orderings (parallel and series) and the two aggregation procedures (AND and OR gates), we denote the corresponding cases as series AND, series OR, parallel AND, and parallel OR.

We use the random variable $Z \in \{0, 1\}$ to denote the aggregated output and first examine the parallel testing protocol with an AND aggregation function. For $n = 2$ parallel tests, the

**Table 1. Examples of parallel and series test protocols that have been used in COVID-19 seroprevalence studies.**

|  | parallel | series |
|---|---|---|
| **AND** | $n = 2$: Slovenia (nationwide) [55] | $n = 2$: Norrbotten County, Sweden [56] |
|  |  | $n = 3$: Vienna, Austria [42] |
| **OR** | $n = 2$: South Africa (three communities) [57] | - |
|  | $n = 4$: Santiago, Chile [43] |  |

 

sensitivity and specificity are

$$\mathrm{TPR}_{1\wedge2}^{(\mathrm{p})} = \Pr(Z = 1 \mid X = 1) = \Pr(Y_1 = 1, Y_2 = 1 \mid X = 1) = \mathrm{TPR}_1\mathrm{TPR}_2 \tag{3}$$

and

$$\begin{aligned}
\mathrm{TNR}_{1\wedge2}^{(\mathrm{p})} &= \Pr(Z = 0 \mid X = 0) \\
&= 1 - \Pr(Y_1 = 1, Y_2 = 1 \mid X = 0) \\
&= \Pr(Y_1 = 0, Y_2 = 0 \mid X = 0) + \Pr(Y_1 = 0, Y_2 = 1 \mid X = 0) \\
&\qquad\qquad + \Pr(Y_1 = 1, Y_2 = 0 \mid X = 0) \\
&= \mathrm{TNR}_1\mathrm{TNR}_2 + \mathrm{TNR}_1(1 - \mathrm{TNR}_2) + \mathrm{TNR}_2(1 - \mathrm{TNR}_1) \\
&= \mathrm{TNR}_1 + (1 - \mathrm{TNR}_1)\mathrm{TNR}_2 \, ,
\end{aligned} \tag{4}$$

respectively. In Eqs (3) and (4), we assumed that the results of different tests are conditionally independent given the disease status. This assumption is commonly made in the medical decision-making literature because it simplifies the mathematical analysis of aggregated test results and aligns with manufacturers' reporting practices, which typically do not report potential dependencies between individual tests. However, test results may actually be correlated or anti-correlated. For instance, consider a SARS-CoV-2 PCR test and an IgG antibody test for the virus' spike protein. The IgG response can take weeks to develop after infection [58], by which time the PCR test is likely to return a negative result since the infection may have cleared. An example where approximate independence might hold is in symptom-based diagnostics that assess different aspects of the same disease. For instance, in some viral infections, the occurrence of fever and loss of taste may show a low degree of dependence, though this can vary depending on the disease and the underlying physiological processes. In the Materials and methods section, we use a dataset containing the test results from nine antibody assays [58] to quantify the level of dependence between them.

The derivations of Eqs (3) and (4) are applicable to $n = 2$ parallel tests, where both the first *and* the second test results must be positive for classifying a sample as positive (an AND gate). If, however, the classification is based on the first *or* the second result being positive (an OR gate), the sensitivity and specificity are

$$\begin{aligned}
\mathrm{TPR}_{1\vee2}^{(\mathrm{p})} &= \Pr(Z = 1 \mid X = 1) \\
&= 1 - \Pr(Y_1 = 0, Y_2 = 0 \mid X = 1) \\
&= \Pr(Y_1 = 1, Y_2 = 1 \mid X = 1) + \Pr(Y_1 = 1, Y_2 = 0 \mid X = 1) \\
&\qquad\qquad + \Pr(Y_1 = 0, Y_2 = 1 \mid X = 1) \\
&= \mathrm{TPR}_1\mathrm{TPR}_2 + \mathrm{TPR}_1(1 - \mathrm{TPR}_2) + \mathrm{TPR}_2(1 - \mathrm{TPR}_1) \\
&= \mathrm{TPR}_1 + (1 - \mathrm{TPR}_1)\mathrm{TPR}_2
\end{aligned} \tag{5}$$

and

$$\mathrm{TNR}_{1\vee2}^{(\mathrm{p})} = \Pr(Z = 0 \mid X = 0) = \Pr(Y_1 = 0, Y_2 = 0 \mid X = 0) = \mathrm{TNR}_1\mathrm{TNR}_2 \, , \tag{6}$$

respectively. As in the derivations for the AND protocol, we assume that test results are independent. An application of the OR protocol might be testing for a disease when two versions of the same disease are circulating (*e.g.*, influenza A and B). In this case, one may aggregate the results of two strain-specific tests to determine if the person has influenza. (*i.e.*, if the first *or* the second test is positive).

 

When results from different tests depend on each other, we show in the Materials and methods section how to use Boole–Fréchet inequalities to formulate tight bounds that relate the sensitivities and specificities of AND and OR aggregations to those of the individual tests.

Given the assumptions in deriving the AND and OR aggregation protocols, we expect that the true positive rate is lower under AND aggregation (since all tests must be positive for a positive diagnosis) and vice-versa that the true negative rate is lower under OR aggregation (since all tests must be negative for a negative diagnosis). Based on Eqs (3) and (6), we obtain

$$\text{TPR}_{1\vee 2}^{(\text{p})} \geq \text{TPR}_{1\wedge 2}^{(\text{p})} \quad \text{and} \quad \text{TNR}_{1\vee 2}^{(\text{p})} \leq \text{TNR}_{1\wedge 2}^{(\text{p})} \tag{7}$$

for all $\text{TPR}_i$ and $\text{TNR}_i$ ($i \in \{1, 2\}$).

Instead of administering two tests in parallel, one may also consider series testing in which whether or not the second test is administered depends on the outcome of the first test. In contrast to parallel testing with an AND aggregation, the second test in the corresponding sequential testing protocol does not have to be performed if the outcome of the first test is negative. The sensitivity and specificity of series testing under AND aggregation are

$$\text{TPR}_{1\wedge 2}^{(\text{s})} = \text{TPR}_1\text{TPR}_2 \quad \text{and} \quad \text{TNR}_{1\wedge 2}^{(\text{s})} = \text{TNR}_1 + (1 - \text{TNR}_1)\text{TNR}_2 \,, \tag{8}$$

respectively. For the corresponding series OR test, we have

$$\text{TPR}_{1\vee 2}^{(\text{s})} = \text{TPR}_1 + (1 - \text{TPR}_1)\text{TPR}_2 \quad \text{and} \quad \text{TNR}_{1\vee 2}^{(\text{s})} = \text{TNR}_1\text{TNR}_2 \,. \tag{9}$$

Notice that the sensitivities and specificities of the aggregated tests are the same regardless of whether a parallel or sequential aggregation protocol is employed. However, in a sequential protocol, fewer tests need to be administered, making this option more economically viable, especially for rapid antigen tests, characterized by lower sensitivity. For tests with extended processing times, such as enzyme-linked immunosorbent assay (ELISA) and RT-PCR tests, one may still prefer parallel test protocols to avoid substantial delays between the first and second tests.

Mathematically, the TPRs and TNRs of the studied combined testing protocols bound the TPRs and TNRs of the constituent tests according to

$$\text{TPR}_{1\wedge 2} \leq \text{TPR}_i \leq \text{TPR}_{1\vee 2} \quad \text{and} \quad \text{TNR}_{1\vee 2} \leq \text{TNR}_i \leq \text{TNR}_{1\wedge 2} \quad \text{for} \quad i \in \{1, 2\} \,. \tag{10}$$

We will show that this bounding result also holds for $n \geq 3$ tests.

**Saving tests with series testing.** To fully cover a population of $N$ individuals using parallel testing would require $2N$ tests. By contrast, series testing involves administering an initial test to all individuals. In the series AND aggregation function, a second test is required if and only if the first test yields a positive result. The probability of this event is $f\text{TPR}_1 + (1 - f)(1 - \text{TNR}_1)$, where $f \in [0, 1]$ is the prevalence, the fraction of the total population carrying a disease. In the series OR aggregation function, a second test is necessary if and only if the first test is negative, and the probability of this event is $f(1 - \text{TPR}_1) + (1 - f)\text{TNR}_1$. Both series testing protocols achieve the same sensitivity and specificity as the parallel test but with fewer tests, specifically, $N(1 + f\text{TPR}_1 + (1 - f)(1 - \text{TNR}_1))$ tests for the series AND function and $N(1 + f(1 - \text{TPR}_1) + (1 - f)\text{TNR}_1)$ tests for the series OR function, instead of $2N$ when conducted in parallel.

We assume that there are enough tests to cover the entire population $N$. The population $N$ is at most half the number of available tests (equal to half under parallel testing). Given the disease prevalence $f$, we now compute the ratio of the number of required tests under parallel testing and the corresponding number of sequential tests. The ratios for the AND and OR

aggregation methods are

$$\left.\frac{\text{parallel tests}}{\text{series tests}}\right|_{1\wedge2} = \frac{2}{1 + f\,\text{TPR}_1 + (1-f)(1-\text{TNR}_1)} \tag{11}$$

and

$$\left.\frac{\text{parallel tests}}{\text{series tests}}\right|_{1\vee2} = \frac{2}{1 + f(1-\text{TPR}_1) + (1-f)\text{TNR}_1}\,, \tag{12}$$

respectively. Both ratios lie between 1 and 2; parallel testing always requires more tests than series testing. Besides the ground truth prevalence $f$, this ratio also depends on the disposition of the first test that determines if a second test is warranted. The first test result in turn, depends on its sensitivity $\text{TPR}_1$ and specificity $\text{TNR}_1$. In Fig 2, we show the ratios (11) and (12) as a function of prevalence $f$ for three different combinations of true positive and true negative rates: (i) $\text{TPR}_1 = 0.95$ and $\text{TNR}_1 = 0.95$, (ii) $\text{TPR}_1 = 0.90$ and $\text{TNR}_1 = 0.95$, and (ii) $\text{TPR}_1 = 0.95$ and $\text{TNR}_1 = 0.90$. When there are no infected individuals in the population (*i.e.*, $f = 0$), the parallel to series ratios are $2/(2-\text{TNR}_1)$ and $2/(1+\text{TNR}_1)$ for the AND and OR aggregation schemes, respectively. If all $N$ individuals in a population are infected (*i.e.*, $f = 1$), the ratios are $2/(1+\text{TPR}_1)$ and $2/(2-\text{TPR}_1)$ for the AND and OR aggregation schemes, respectively.

It is also straightforward to show that for $f < f_c$, where $f_c$ is the critical prevalence defined as

$$f_c = \frac{2\text{TNR}_1 - 1}{2(\text{TPR}_1 + \text{TNR}_1 - 1)}\,, \tag{13}$$

the number of required AND-aggregated series tests is less than the number of required OR-aggregated series tests. Equivalently, for $f < f_c$, the curve representing the ratio of the required parallel-to-series tests under the AND protocol given in Eq (11) falls above the corresponding OR protocol curve given in Eq (12). The trends observed in the parallel-to-series ratios as a

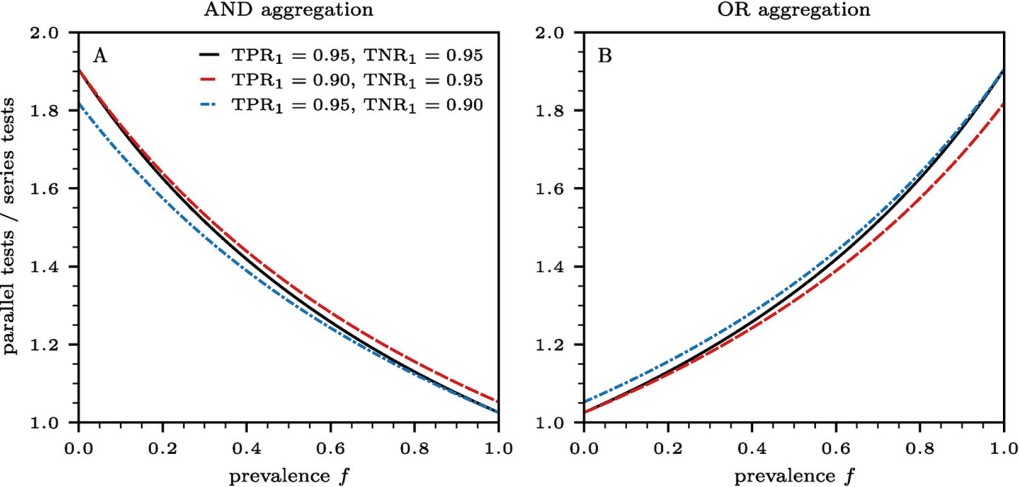

**Fig 2. The ratio of the number of parallel tests to the number of series tests necessary to determine the aggregated output from $n = 2$ tests as a function of prevalence $f$.** Results in panels (A) and (B) are based on AND and OR aggregations of two tests, using Eqs (11) and (12), respectively. We consider three different combinations of true positive and true negative rates (solid black lines: $\text{TNR}_1 = 0.95$ and $\text{TNR}_1 = 0.95$; dashed red lines: $\text{TNR}_1 = 0.90$ and $\text{TNR}_1 = 0.95$; dash-dotted blue lines: $\text{TNR}_1 = 0.95$ and $\text{TNR}_1 = 0.90$). The critical values $f_c$ for which the ratios in panel (A) are larger than the ratios in panel (B) are given, respectively, by $f_c = 0.50, 0.47, 0.53$. For $f < f_c$ greater savings are achieved by utilizing the AND-aggregated series tests, compared to the OR-aggregated series test.

function of prevalence $f$, shown in Fig 2A, confirm that AND aggregation yields greater test savings through series testing for prevalences $f < f_c$. In contrast, OR aggregation results in larger savings for $f > f_c$, as illustrated in Fig 2B.

According to Eq (13), the quantity $f_c$ is meaningful when $TNR_1 \geq 1/2$ and when $TPR_1 + TNR_1 > 1$. The latter condition implies that the true positive rate of the first test is greater than its false positive rate (*i.e.*, $1 - TNR_1$). A test that satisfies this condition is said to have "discriminatory power" [38]. Typical values of $TPR_1$ and $TNR_1$ yield intermediate values of $f_c \approx 0.5$ as shown in Fig 2. Another scenario in which $f_c$ is mathematically meaningful is when $TNR_{1 \leq 1/2}$ and $TPR_1 + TNR_1 < 1$. In this case, the trends in Fig 2A and 2B are reversed compared to the ones just discussed. This scenario, however, is highly unrealistic, as the first test is misleading since its false positive rate is greater that its true positive rate.

**Positive predictive value.**   Measures such as sensitivity and specificity fail to appropriately take into account the prevalence of a disease $f$ [59]. In this context, a more appropriate measure is the positive predictive value (PPV), also known as precision, defined as

$$PPV = \frac{f\,TPR}{f\,TPR + (1-f)(1-TNR)} \ . \tag{14}$$

The PPV is the number of true positives divided by the number of positive calls. Similarly, the negative predictive value (NPV) is the number of true negatives divided by the number of negative calls. That is,

$$NPV = \frac{(1-f)\,TNR}{(1-f)\,TNR + f(1-TPR)} \ . \tag{15}$$

Here, TPR and TNR represent the overall true positive and true negative rates of the aggregate testing protocol. By defining the utility gain associated with treating a sick individual and the utility loss associated with treating a healthy individual, it is possible to establish a relationship between PPV, NPV, and the treatment threshold. This threshold is the point where the expected treatment gain equals the expected treatment loss [38].

Based on Eqs (14) and (15), one can show that the PPV is an increasing function of $f$ and that the NPV is a decreasing function of $f$. These equations also yield $PPV \geq NPV$ when

$$f \geq \frac{\sqrt{TNR(1-TNR)}}{\sqrt{TNR(1-TNR)} + \sqrt{TPR(1-TPR)}} \ . \tag{16}$$

For multiple tests, the PPV and NPV are independent of the test ordering (parallel or series); however, they depend on the TPRs and TNRs of the individual tests in different ways depending whether the AND or the OR aggregation protocol is used. Specifically, we have

$$PPV_{1 \wedge 2} \geq PPV_{1 \vee 2} \,, \qquad \forall f \tag{17}$$

if

$$\frac{TPR_1 + TPR_2}{TPR_1 TPR_2} \leq \frac{(1 - TNR_1) + (1 - TNR_2)}{(1 - TNR_1)(1 - TNR_2)} \ . \tag{18}$$

Similarly, we find

$$NPV_{1 \wedge 2} \leq NPV_{1 \vee 2} \,, \qquad \forall f \tag{19}$$

if

$$\frac{\mathrm{TNR}_1 + \mathrm{TNR}_2}{\mathrm{TNR}_1 \mathrm{TNR}_2} \leq \frac{(1 - \mathrm{TPR}_1) + (1 - \mathrm{TPR}_2)}{(1 - \mathrm{TPR}_1)(1 - \mathrm{TPR}_2)}. \tag{20}$$

The conditions in Eqs (18) and (20) are always satisfied if tests with discriminatory power are used, *i.e.* if $\mathrm{TPR}_1 + \mathrm{TNR}_1 \geq 1$ and $\mathrm{TPR}_2 + \mathrm{TNR}_2 \geq 1$.

In Fig 3, we show the dependence of PPV and NPV on the prevalence $f$. We used sensitivities and specificities associated with AND and OR aggregations for two tests [see Eqs (3) – (6)]. We also include the corresponding PPV and NPV of individual (unaggregated) tests for reference. Tests aggregated with the AND function yield substantially higher PPVs compared to those aggregated with an OR function for all $f$, while the OR aggregation results in notably higher NPVs than those obtained using the AND aggregation. For low prevalence $f$, a good

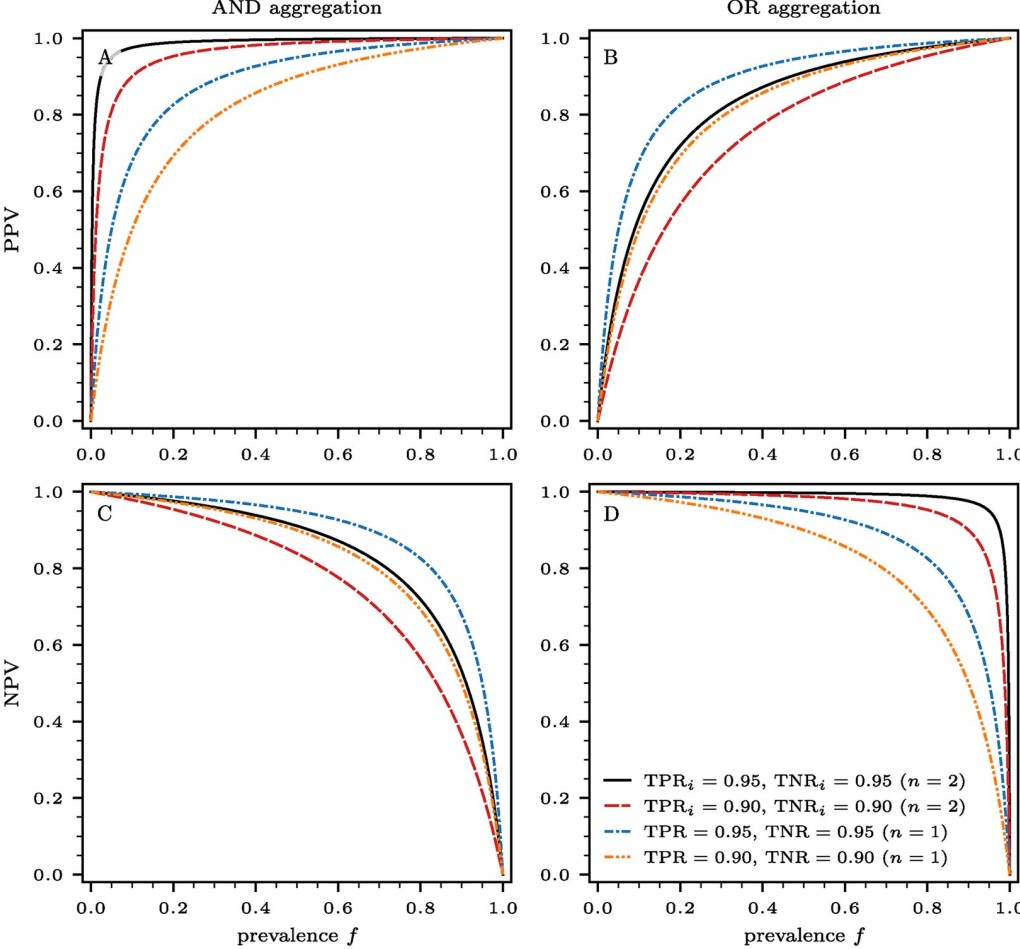

**Fig 3. Positive predictive value (PPV) and negative predictive value (NPV) as a function of prevalence $f$.** The results that we show in panels (A,C) and (B,D) are based on AND and OR aggregations of $n = 2$ tests, using Eqs (14) and (15), respectively. We denote the sensitivities and specificities of the two tests $i \in \{1, 2\}$ by $\mathrm{TNR}_i$ and $\mathrm{TNR}_i$, respectively. We consider two different combinations of true positive and true negative rates (solid black lines: $\mathrm{TNR}_i = 0.95$ and $\mathrm{TNR}_i = 0.95$; dashed red lines: $\mathrm{TNR}_i = 0.90$ and $\mathrm{TNR}_i = 0.90$). As a reference, we also show results for single tests without further aggregation (dash-dotted blue line: $\mathrm{TNR} = 0.95$ and $\mathrm{TNR} = 0.95$; dash-dot-dotted orange line: $\mathrm{TNR} = 0.90$ and $\mathrm{TNR} = 0.90$). These curves are independent of the ordering (parallel or series) method used.

PPV-NPV tradeoff is obtained under AND aggregation, whereas OR aggregation is best for high prevalence $f$. For low $f$, the main source of test error is the false positive rate $1 - TNR$. This term is minimized under the AND aggregation as per Eq (10). Similarly, for high $f$, the primary source of test error is the false negative rate $1 - TPR$, which is minimized under the OR aggregation as per in Eq (10).

So far, we have shown that when tests with discriminatory power are used for diseases with prevalence $f < f_c$, the AND aggregation protocol leads to the greatest reduction in the number of required tests when applied in series. Additionally, the AND aggregation protocol leads to larger PPV values compared to the OR protocol. Conversely, the potential savings in the number of required tests under the OR aggregation protocol are smaller for $f < f_c$ and the NPV is larger than under the AND protocol. Thus, our analysis suggests that for $n = 2$ tests, the most suitable protocol for minimizing test usage and maximizing the PPV estimate in low-prevalence scenarios is the series AND method.

To provide further analytical insight into the properties of repeated tests, we consider aggregation functions involving more than two tests in the next section.

### Aggregating results from more than two tests

In Table 1, we list examples of SARS-CoV-2 seroprevalence studies where up to four tests were administered using various combinations of parallel and series ordering with AND and OR aggregation [43]. These examples illustrate the use of various testing configurations (*i.e.*, different Boolean functions, varying numbers of tests, and both series and parallel processing) during the SARS-CoV-2 pandemic. While the results of these different tests may show dependencies, we proceed with our derivations under the assumption that the results of individual tests are conditionally independent given the disease status. If the results of individual tests exhibit dependence effects, bounds relating the sensitivities and specificities of several aggregation functions to those of the individual tests can be derived using the Boole–Fréchet inequalities [60–63] (see Materials and methods for further details).

For $n = 3$ tests, there are $r = 2^3 = 8$ possible output sequences and $m = 2^r = 2^8 = 256$ possible input-output mappings. For $n = 4$, these numbers increase to $r = 2^4 = 16$ and $m = 2^r = 2^{16} = 65,536$ respectively. Given the large number of possible ways of combining $n$ tests, we will derive sensitivities and specificities for a few select choices and otherwise resort to an algorithmic evaluation of test performances as detailed in the following section.

Eqs (3)–(8) show that for $n = 2$, parallel and series test protocols carry the same sensitivities and specificities. This equivalence remains valid for $n \geq 3$ tests, so for notational simplicity we suppress the "s" and "p" superscripts that distinguish them.

For $n = 3$ tests and an AND aggregation, the sensitivity and specificity are

$$TPR_{1 \wedge 2 \wedge 3} = TPR_1 TPR_2 TPR_3 \tag{21}$$

and

$$\begin{aligned} TNR_{1 \wedge 2 \wedge 3} = \ & TNR_1 + TNR_2 + TNR_3 - TNR_1 TNR_2 \\ & - TNR_1 TNR_3 - TNR_2 TNR_3 + TNR_1 TNR_2 TNR_3, \end{aligned} \tag{22}$$

respectively. Similarly, the sensitivity and specificity of an R test protocol with $n = 3$ tests are

$$\begin{aligned} TPR_{1 \vee 2 \vee 3} = \ & TPR_1 + TPR_2 + TPR_3 - TPR_1 TPR_2 \\ & - TPR_1 TPR_3 - TPR_2 TPR_3 + TPR_1 TPR_2 TPR_3 \end{aligned} \tag{23}$$

and

$$\mathrm{TNR}_{1\vee2\vee3} = \mathrm{TNR}_1\,\mathrm{TNR}_2\,\mathrm{TNR}_3\,. \tag{24}$$

The overall sensitivity and specificity of the limiting AND and OR aggregations for general $n$-tests are

$$\mathrm{TPR}_n^{\mathrm{AND}} = \prod_{i=1}^{n} \mathrm{TPR}_i\,, \quad \mathrm{TNR}_n^{\mathrm{AND}} = 1 - \prod_{i=1}^{n} \left(1 - \mathrm{TNR}_i\right), \tag{25}$$

and

$$\mathrm{TPR}_n^{\mathrm{OR}} = 1 - \prod_{i=1}^{n} \left(1 - \mathrm{TPR}_i\right), \quad \mathrm{TNR}_n^{\mathrm{OR}} = \prod_{i=1}^{n} \mathrm{TNR}_i\,, \tag{26}$$

where we assumed that the results of different tests are conditionally independent given the disease status.

In line with Eq (10), the TPRs and TNRs of the combined testing protocols satisfy

$$\mathrm{TPR}_n^{\mathrm{AND}} \leq \mathrm{TPR}_i \leq \mathrm{TPR}_n^{\mathrm{OR}} \quad \text{and} \quad \mathrm{TNR}_n^{\mathrm{OR}} \leq \mathrm{TNR}_i \leq \mathrm{TNR}_n^{\mathrm{AND}} \quad \forall i \in \{1, \cdots, n\}\,. \tag{27}$$

For odd $n \geq 3$, one can also employ a majority aggregation, where at least $(n + 1)/2$ tests have to be positive for the combined test to be positive. The majority function is intermediate relative to the "all" and "any" characteristics of the AND and OR functions, respectively. The sensitivity of a majority aggregation of $n = 3$ tests is

$$\mathrm{TPR}_{\mathrm{M}(1,2,3)} = \mathrm{TPR}_1\,\mathrm{TPR}_2 + \mathrm{TPR}_1\,\mathrm{TPR}_3 + \mathrm{TPR}_2\,\mathrm{TPR}_3 - 2\mathrm{TPR}_1\,\mathrm{TPR}_2\,\mathrm{TPR}_3\,, \tag{28}$$

and the corresponding specificity is

$$\mathrm{TNR}_{\mathrm{M}(1,2,3)} = \mathrm{TNR}_1\,\mathrm{TNR}_2 + \mathrm{TNR}_1\,\mathrm{TNR}_3 + \mathrm{TNR}_2\,\mathrm{TNR}_3 - 2\mathrm{TNR}_1\,\mathrm{TNR}_2\,\mathrm{TNR}_3\,. \tag{29}$$

Because the majority function interpolates between the extremes of requiring all tests to be positive (AND) and requiring at least just one positive result (OR), the quantities $\mathrm{TPR}_{\mathrm{M}(1,2,3)}$ and $\mathrm{TNR}_{\mathrm{M}(1,2,3)}$ are bounded by the sensitivities and specificities of the AND and OR aggregations according to

$$\mathrm{TPR}_{1\wedge2\wedge3} \leq \mathrm{TPR}_{\mathrm{M}(1,2,3)} \leq \mathrm{TPR}_{1\vee2\vee3} \quad \text{and} \quad \mathrm{TNR}_{1\vee2\vee3} \leq \mathrm{TNR}_{\mathrm{M}(1,2,3)} \leq \mathrm{TNR}_{1\wedge2\wedge3}\,. \tag{30}$$

In Fig 4, we show receiver operating characteristic (ROC) curves for various combinations of tests and aggregation functions. In Fig 4A, we present the sensitivities and false positive rates for AND and OR aggregations with $n = 2$ tests. Additionally, in Fig 4B, we consider AND, OR, and majority aggregation for $n = 3$ tests. Under AND aggregation, the sensitivities and false positive rates of the combined tests are lower than those of the individual tests. The opposite holds for OR aggregation. These findings are in agreement with the analytical results in Eq (27). Finally, when examining $n = 3$ tests, the majority function yields greater sensitivities and reduced false positive rates than the individual isolated tests. The error bars in both panels represent the bounds defined by the Boole–Fréchet inequalities (see Materials and methods).

## Efficiently combining $n$ tests

For a given set of $n$ tests, what aggregation protocols yield the best sensitivities and specificities? As discussed in the prior sections, there exist numerous possibilities to combine individual tests, and the mathematical expressions for aggregated sensitivities and specificities can be

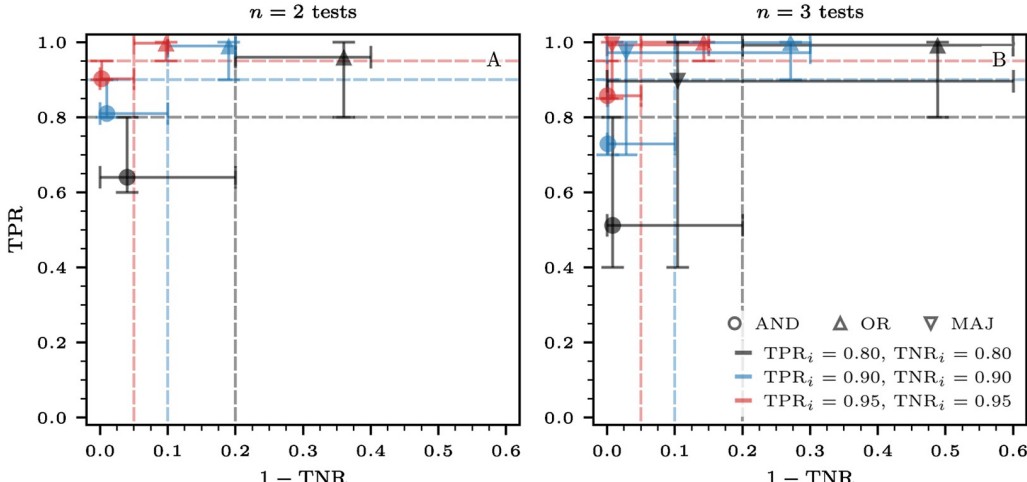

**Fig 4. Receiver operating characteristic (ROC) curves for various combinations of tests and aggregation functions.**
(A) We consider $n = 2$ tests and two distinct aggregation functions (disks: AND aggregation; triangles: OR aggregation).
(B) We consider $n = 3$ tests and the same aggregation functions as in panel (A) along with the majority function
represented by inverted triangles. Markers in black, blue, and red represent combined tests where the underlying tests $i \in$
$\{1, \ldots, n\}$ have sensitivities (TPR$_i$) and specificities (TNR$_i$) set to 0.8, 0.9, and 0.95, respectively. Dashed lines indicate the
sensitivities and false positive rates (*i.e.*, 1 − TNR) of the individual isolated tests. Under AND aggregation, both the
sensitivities and false positive rates of the combined tests are smaller than those of the individual tests. The opposite holds
for OR aggregation. When considering $n = 3$ tests, the majority function results in higher sensitivities and smaller false
positive rates compared to the individual isolated tests. This function provides a tradeoff between the "all" and "any"
characteristics of AND and OR aggregations. The results shown are independent of the ordering (parallel or series)
method used. The error bars in both panels represent the bounds defined by the Boole–Fréchet inequalities
(see Materials and methods), which apply irrespective of the dependence structure relating the individual tests.

quite lengthy. Therefore, we adopt an algorithmic approach to compute ROC curves associated
with $n$ tests, each potentially having distinct sensitivities and specificities. In this context, we
use the term "efficient test" to refer to an individual or combined test that lies on the ROC
frontier (*i.e.*, the ROC convex hull).

**Aggregation and optimization.** Algorithm 1 shown below computes the most efficient
combination of $n$ conditionally independent tests for given TPR$_i$ and TNR$_i$ of each test $i \in \{1,$
$\ldots, n\}$. The following example illustrates our algorithm. We define $\mathcal{P} = \{P_1, \ldots, P_r\}$ as the set
of possible ordered outcomes deriving from the administration of $n$ tests, where $r = 2^n$. For
example, for $n = 2$ tests, there are $r = 2^n = 2^2 = 4$ permutations and $P_1 = (+, +)$, $P_2 = (+, -)$, $P_3 =$
$(-, +)$, and $P_4 = (-, -)$. Thus, we have

$$\mathcal{P} = \{(+, +), (+, -), (-, +), (-, -)\}. \tag{31}$$

Each of these four outcomes can be mapped to either a positive (+) or negative (−) assigned
disease status. Hence, there are $m = 2^r = 2^4 = 16$ mappings in total. For example, the output
sequence $\mathcal{S} = (+, -, -, -)$ means that only the input $P_1 = (+, +)$ is mapped to an aggregated
"+", and the other permutations $P_2 = (+, -)$, $P_3 = (-, +)$, and $P_4 = (-, -)$ are mapped to "−.".
This case corresponds to the AND aggregation protocol. Similarly, $\mathcal{S} = (+, +, +, -)$ corre-
sponds to the OR aggregation protocol.

We define the sensitivity TPR$_\mathcal{S}$ associated with the output sequence $\mathcal{S} = (S_1, \ldots, S_r)$ in two
steps. First, we define TPR$_\mathcal{S}$ as the sum over the sensitivities TPR$_{S_j}$ ($j \in \{1, \ldots, r\}$) associated

with elements $S_j$ of $\mathcal{S}$. That is,

$$\text{TPR}_{\mathcal{S}} = \sum_{j=1}^{r} \text{TPR}_{S_j} . \tag{32}$$

Second, we define $\text{TPR}_{S_j}$ as follows. If element $S_j$ is "−" (*i.e.*, if the input state $P_j$ gets classified as negative), then we pose $\text{TPR}_{S_j} = 0$. Otherwise, if element $S_j$ is "+", we calculate products of $\text{TPR}_i$ and $1 - \text{TPR}_i$ depending on whether the result from test $i \in \{1, \ldots, n\}$ is positive or negative. That is,

$$\text{TPR}_{S_j} = \begin{cases} \prod_{i=1}^{n} [\text{TPR}_i \delta_{i,+} + (1 - \text{TPR}_i) \delta_{i,-}] , & \text{if } S_j \text{ is } + \\ 0 , & \text{if } S_j \text{ is } - \end{cases}, \tag{33}$$

where $\delta_{i,+} = 1$ if test $i$ is positive and 0 otherwise. Likewise, $\delta_{i,-} = 1$ if test $i$ is negative and 0 otherwise. The product form of Eq (33) arises from the assumption of conditionally independent tests.

We provide a simple example to allow easier interpretation of Eqs (32) and (33). For $n = 2$ tests and $\mathcal{S} = (+, -, -, -)$, Eqs (32) and (33) reduce to the AND aggregation result $\text{TPR}_{1 \wedge 2}$ given in Eqs (3) and (8). Similarly, for $n = 2$ tests and $\mathcal{S} = (+, +, +, -)$, Eq (33) reduces to the OR aggregation result $\text{TPR}_{1 \vee 2}$ given in Eqs (5) and (9).

We follow the same steps to define the specificity $\text{TNR}_{\mathcal{S}}$ of the output sequence $\mathcal{S}$ so that

$$\text{TNR}_{\mathcal{S}} = \sum_{j=1}^{r} \text{TNR}_{S_j} \tag{34}$$

where

$$\text{TNR}_{S_j} = \begin{cases} 0 , & \text{if } S_j \text{ is } + \\ \prod_{i=1}^{n} [(1 - \text{TNR}_i) \delta_{i,+} + \text{TNR}_i \delta_{i,-}] , & \text{if } S_j \text{ is } - \end{cases}. \tag{35}$$

For $n = 2$ tests and $\mathcal{S} = (+, -, -, -)$, Eq (35) reduces to the AND aggregation $\text{TNR}_{1 \wedge 2}$ result given in Eqs (4) and (8). Similarly, for $n = 2$ tests and $\mathcal{S} = (+, +, +, -)$, Eq (35) simplifies to the OR aggregation $\text{TNR}_{1 \vee 2}$ result given in Eqs (6) and (9).

We identify two limiting cases. One is the output sequence $\mathcal{S} = (+, +, +, +)$ where all input permutations $P_j$ are mapped to "+" outcomes. In this case, the aggregated sensitivity and specificity are $\text{TPR}_{\mathcal{S}} = 1$ and $\text{TNR}_{\mathcal{S}} = 0$, respectively. The other limit is the output sequence $\mathcal{S} = (-, -, -, -)$ where all input permutations $P_j$ are mapped to "−" outcomes. Here, the aggregated sensitivity and specificity are $\text{TPR}_{\mathcal{S}} = 0$ and $\text{TNR}_{\mathcal{S}} = 1$, respectively.

Once we have determined all pairs $(\text{TPR}_{\mathcal{S}}, \text{TNR}_{\mathcal{S}})$ associated with the $m = 2^r$ test aggregations, we identify the most efficient test combinations, *i.e.*, those combinations where the underlying sensitivity-specificity pairs reach the highest values. This is achieved by employing a convex-hull algorithm, such as Graham scan [64] and Quickhull [65, 66], to determine the ROC frontier in the $(\text{TPR}_{\mathcal{S}}, 1 - \text{TNR}_{\mathcal{S}})$ space (*i.e.*, true positive-false positive rate space). Instead of manually comparing $\text{TPR}_{\mathcal{S}}$ and $\text{TNR}_{\mathcal{S}}$ values for $m = 2^r$ tests, a convex-hull algorithm can efficiently perform this task and identify the individual or combined tests on the ROC frontier.

We summarize all steps of our algorithm in `Python` pseudocode in Algorithm 1.

**Algorithm 1** Compute the most efficient combinations of $n$ conditionally independent tests.

```
 1: Inputs:
       n, TPRs, TNRs, ConvexHull()
 2: Outputs:
       roc_frontier
 3: input_permutations ← list(itertools.product([0, 1], repeat=n))
       ▷ Generate input permutations 𝒫
 4: input_output_mappings ← list(itertools.product([0, 1],
    repeat = 2ⁿ))   ▷ Generate output sequences 𝒮
 5: TPR_arr ← []
 6: TNR_arr ← []
 7: for input_output_map in input_output_mappings do
 8:   TPR_combined ← []
 9:   TNR_combined ← []
10:   for perm, output_value in zip(input_permutations,
      input_output_map) do
11:     if output_value then
12:       TPR_combined.append($\prod_{i=0}^{n-1}$ (TPRs[i] if perm[i] else 1-TPRs[i]))
      ▷ see Eq (33)
13:     else
14:       TNR_combined.append($\prod_{i=0}^{n-1}$ (1-TNRs[i] if perm[i] else TNRs
          [i]))   ▷ see Eq (35)
15:     end if
16:   end for
17:   TPR_arr.append(sum(TPR_combined))      ▷ Compute aggregated sen-
      sitivity using Eq (33)
18:   TNR_arr.append(sum(TNR_combined))      ▷ Compute aggregated speci-
      ficity using Eq (34)
19: end for
20: points ← concatenate(1-TNR_arr, TPR_arr)
21: convex_hull ← ConvexHull(points)
22: roc_frontier ← []
23: for edge in convex_hull do
24:   if (points[edge, 1][0] ≥ points[edge, 0][0]) and (points[edge,
      1][1] ≥ points[edge, 0][1]) then
25:     roc_frontier.append(points[edge])        ▷ ROC points must sat-
      isfy TPR ≥ 1-TNR
26:   end if
27: end for
28:   return roc_frontier
```

**An example with three antigen tests.** We now show how Algorithm 1 can identify effi-
cient aggregation protocols, using an example with $n = 3$ tests. The sensitivities and specifici-
ties are based on commonly used SARS-CoV-2 antigentests [1]. We list their median
sensitivities and specificities along with their 95% confidence intervals (CIs) in Table 2. How-
ever, depending on the characteristics of these tests, the assumption of conditional indepen-
dence of test results given disease status may not hold.

**Table 2. Median sensitivities and specificities of three commonly used SARS-CoV-2 antigen tests that are based
on studies involving symptomatic patients [1].** Numbers in parentheses denote 95% CIs.

|  | sensitivity | specificity |
|---|---|---|
| **Abbott—Panbio COVID-19 Ag** | 74.8% (67.6—80.8%) | 99.7% (99.6—99.8%) |
| **Innova Medical Group—Innova SARS-CoV-2 Ag** | 68.1% (47.2—83.6%) | 99.0% (98.5—99.3%) |
| **Siemens—CLINITEST Rapid COVID-19 Ag** | 68.7% (48.0—83.8%) | 100% (98.0—100%) |

For $n = 3$ tests, there are $r = 2^n = 2^3 = 8$ permutations of test results and $m = 2^r = 256$ possible input-output mappings. The set of permutations is

$$\mathcal{P} = \{(+,+,+),(+,+,-),(+,-,+),(+,-,-),(-,+,+),(-,+,-),(-,-,+),(-,-,-)\}, \quad (36)$$

and the corresponding output sequence is $\mathcal{S} = (S_1, \ldots, S_r)$, where $r = 8$ and $S_j \in \{+, -\}$. To make the notation simpler, we introduce the Boolean variable $Y_i \in \{0, 1\}$ for each test $i \in \{1, \ldots, 3\}$ and map $\mathcal{S}$ to its corresponding Boolean expression.

We do this by first using the median sensitivities and specificities of the three tests from Table 2 as inputs in Algorithm 1 for various aggregation protocols, which are then used to derive the corresponding ROC curve shown in Fig 5A. On this curve, there exist two extreme cases: (i) an aggregation method where both sensitivity $\text{TPR}_\mathcal{S}$ and false positive rate $1 - \text{TNR}_\mathcal{S}$ are equal to 0, effectively classifying all input sequences as negative. This corresponds to $S_j = -$ for all $j \in \{1, \ldots, 8\}$; and (ii) an aggregation method with sensitivity $\text{TPR}_\mathcal{S}$ and false positive rate $1 - \text{TNR}_\mathcal{S}$ both at 1, resulting in the classification of all input sequences as positive, corresponding to $S_j = +$ for all $j \in \{1, \ldots, 8\}$.

We also include four more aggregation methods on the ROC curve shown in Fig 5A. The first of these requires that only the last test (*i.e.*, Siemens) is positive, irrespective of the outcomes of the other two. This aggregation method corresponds to $\mathcal{S} = (+,-,+,-,+,-,+,-)$ and is denoted $Y_3$. It exhibits the smallest possible false positive rate, $1 - \text{TNR}_\mathcal{S} = 0$, which is intuitive given that the Siemens test also has the lowest median false positive rate of 0. Its sensitivity is $\text{TPR}_\mathcal{S} = 68.7\%$.

The next combined test shown on the ROC curve requires both the first and the second tests (*i.e.*, Abbott and Innova), or the last one (*i.e.*, Siemens) to yield positive results. This protocol corresponds to $\mathcal{S} = (+,+,+,-,+,-,+,-)$ and can be written in Boolean algebra as $(Y_1 \wedge Y_2) \vee Y_3$. Using the values listed in Table 2 and Eqs (33) and (35), it can be verified that its sensitivity and false positive rate are $\text{TPR}_\mathcal{S} = 84.6\%$ and $1 - \text{TNR}_\mathcal{S} = 3.0 \times 10^{-3}\%$, respectively.

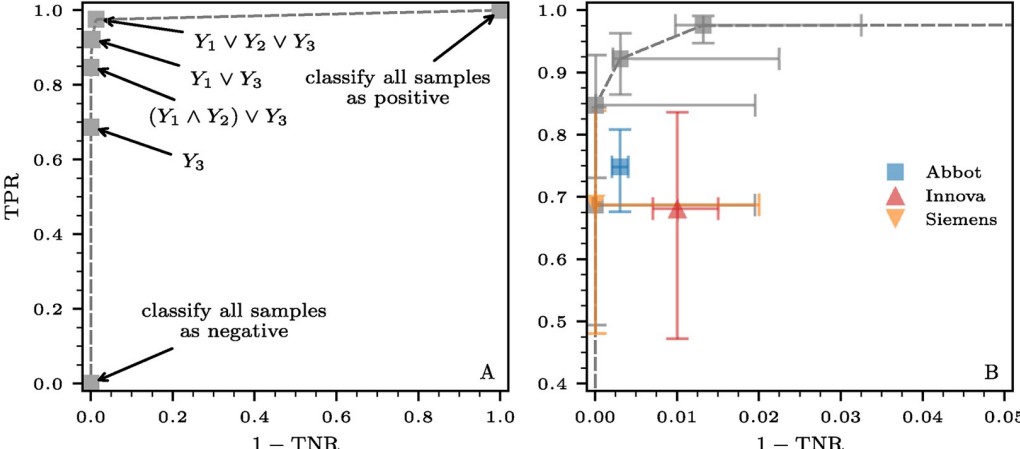

**Fig 5. ROC curves associated with the aggregation of three antigen tests (Abbot, Innova, and Siemens).** The sensitivities and specificities of the $n = 3$ tests are listed in Table 2. (A) The ROC curve associated with the aggregation of the three antigen tests as derived from Eqs (33) and (35). We use $Y_i \in \{0, 1\}$ to denote the outcome of test $i \in \{1, 2, 3\}$. The dashed curve is a visual guide connecting the tests on the ROC curve. (B) A magnified view of the ROC curve without the trivial combined tests that classify all samples as either negative or positive. The error bars indicate the 95% CIs that we generated from $10^6$ samples of beta distributions capturing the 95% CIs of the underlying individual sensitivities and specificities.

We can improve the aggregated sensitivity by omitting the second test (*i.e.*, Innova), which has the lowest sensitivity at 68.1%; the tradeoff is to accept a slightly higher false positive rate. This protocol yields the next point on the ROC curve. It corresponds to $\mathcal{S} = \{+, +, +, +, +, -, +, -\}$ and can be written using an OR aggregation over the first and last tests, *i.e.*, $Y_1 \vee Y_3$. Eqs (33) and (35) yield the sensitivity $\text{TPR}_{\mathcal{S}} = 92.1\%$ and the false positive rate $1 - \text{TNR}_{\mathcal{S}} = 0.3\%$.

Finally, the largest sensitivity smaller than 100% is achieved through an OR aggregation over all tests *i.e.*, for $Y_1 \vee Y_2 \vee Y_3$. This corresponds to $\mathcal{S} = \{+, +, +, +, +, +.+, -\}$, an output sequence with sensitivity $\text{TPR}_{\mathcal{S}} = 97.5\%$ and false positive rate $1 - \text{TNR}_{\mathcal{S}} = 1.3\%$ as per Eqs (33) and (35).

For a more detailed comparison between combined and individual tests, we show a magnified view of the four non-trivial aggregations in the ROC frontier in Fig 5B and include individual tests. In this plot, we incorporate CIs alongside median sensitivities and false positive rates. We generate these CIs from $10^5$ samples of beta distributions capturing the 95% CIs of the underlying individual sensitivities and specificities (see Materials and methods for further details). We observe that the two OR protocols, $Y_1 \vee Y_3$ and $Y_1 \vee Y_2 \vee Y_3$, exhibit significantly higher sensitivity compared to each individual test.

## Estimating prevalence

In the preceding sections, we have described how repeating and aggregating test results can substantially enhance sensitivity and specificity. This enhancement can contribute to improved infectious-disease surveillance and management [45–48] by providing more accurate estimates $\hat{f}$ of the true prevalence $f$ in a population. The prevalences $\hat{f}$ and $f$ may be time-dependent and stratified, *e.g.*, according to age, (*i.e.*, $\hat{f} \equiv \hat{f}(a_k, t)$ and $f \equiv f(a_k, t)$ where $a_k$ is a given age). Additionally, depending on the test type and the disease being considered, it may be necessary to account for time-dependent sensitivities and specificities. For instance, in the context of SARS-CoV-2 tests, antibody waning is known to affect test characteristics over time [51]. In the examples we consider in this work, we take sensitivities and specificities to remain constant.

**Correcting test errors.** We distinguish between two types of prevalences: (i) the measured prevalence, $\hat{f}^*_{\mathcal{S}}(a_k, t)$, which is derived from testing a sample of the population using the aggregation method with output sequence $\mathcal{S}$, and (ii) the measured, error-corrected prevalence, $\hat{f}(a_k, t)$, which is an estimate of the true disease prevalence, $f(a_k, t)$. If we also assume that the selected sample is unbiased and representative of the infection behavior in the entire population, we can identify the estimate $\hat{f}(a_k, t)$ with the actual prevalence $f(a_k, t)$ and write $\hat{f}(a_k, t) = f(a_k, t)$. For $n$ combined tests with output sequence $\mathcal{S}$, the quantities $\hat{f}^*_{\mathcal{S}}(a_k, t)$ and $\hat{f}(a_k, t)$ are related via

$$\hat{f}^*_{\mathcal{S}}(a_k, t) = \hat{f}(a_k, t)\text{TPR}_{\mathcal{S}} + (1 - \hat{f}(a_k, t))(1 - \text{TNR}_{\mathcal{S}}), \tag{37}$$

which yields

$$\hat{f}(a_k, t) = \frac{\hat{f}^*_{\mathcal{S}}(a_k, t) + \text{TNR}_{\mathcal{S}} - 1}{\text{TPR}_{\mathcal{S}} + \text{TNR}_{\mathcal{S}} - 1}, \tag{38}$$

a generalized Rogan–Gladen prevalence estimate [44] that accounts for the sensitivity and specificity of the combined tests with output sequence $\mathcal{S}$. We omit the subscript $\mathcal{S}$ in $\hat{f}(a_k, t)$ since the error-corrected prevalence is an estimate of the true prevalence and should not

depend of the method used for aggregating test results. For example, for $n = 2$ tests under AND and OR aggregation and using Eqs (3)–(8), we have

$$\hat{f}(a_k, t) = \frac{\hat{f}^*_{1\wedge2}(a_k, t) + \text{TNR}_1 + \text{TNR}_2 - \text{TNR}_1\text{TNR}_2 - 1}{\text{TPR}_1\text{TPR}_2 + \text{TNR}_1 + \text{TNR}_2 - \text{TNR}_1\text{TNR}_2 - 1}, \quad (39)$$

and

$$\hat{f}(a_k, t) = \frac{\hat{f}^*_{1\vee2}(a_k, t) + \text{TNR}_1\text{TNR}_2 - 1}{\text{TPR}_1 + \text{TPR}_2 - \text{TPR}_1\text{TPR}_2 + \text{TNR}_1\text{TNR}_2 - 1}, \quad (40)$$

respectively.

In Fig 6, we show the measured (uncorrected) prevalences $\hat{f}^*_{1\wedge2}$ and $\hat{f}^*_{1\vee2}$ associated with the AND and OR aggregations using Eq (37) and Eqs (3)–(8) for the corresponding $\text{TPR}_S$ and $\text{TNR}_S$, and using different sensitivities and specificities for $n = 2$ tests. For simplicity, we assume that samples are unbiased and that the measured, error-corrected prevalence $\hat{f}$ can be identified with the true prevalence $f$.

In line with our findings regarding PPV and NPV and the trends shown in Fig 3, we observe in Fig 6 that $\hat{f}^*_{1\wedge2}$ deviates only slightly from the true prevalence $f$ when the true prevalence is low, whereas under OR aggregation $\hat{f}^*_{1\vee2}$ accurately approximates $f$ when the true prevalence is high.

As a real-world example of prevalence correction under aggregated testing, we consider the seroprevalence study from Norrbotten, Sweden (May 25—June 5, 2020) [56]. In this study, two SARS-CoV-2 tests, the Abbott SARS-CoV-2 IgG kit and the Euroimmun Anti-SARS-CoV-2 ELISA (IgG), were administered to an age-stratified population and combined using an AND function. Our primary goal in this example is to use it as a proof of concept to demonstrate how errors from combined tests can be corrected in the corresponding prevalence

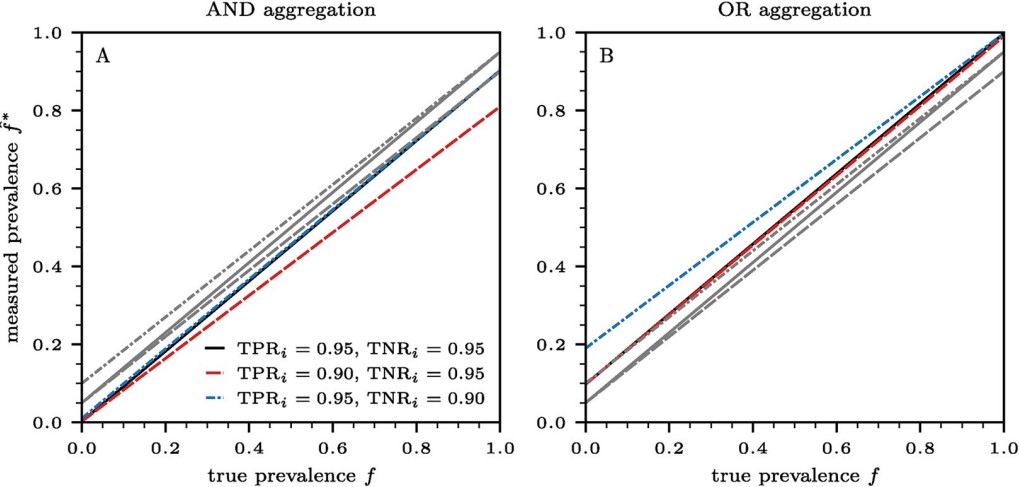

**Fig 6. Measured prevalence $\hat{f}^*$ as a function of true prevalence $f$ under the assumption that the measured, error-corrected prevalence $f$ in Eq (37) can be identified with the true prevalence $f$.** The results shown in panels (A) and (B) are based on AND and OR aggregations of two tests $i \in \{1, 2\}$, respectively. We consider three different combinations of true positive and true negative rates (solid black lines: $\text{TNR}_i = 0.95$ and $\text{TNR}_i = 0.95$; dashed red lines: $\text{TNR}_i = 0.90$ and $\text{TNR}_i = 0.95$; dash-dotted blue lines: $\text{TNR}_i = 0.95$ and $\text{TNR}_i = 0.90$). Grey lines indicate measured prevalences associated with individual tests.

estimates. However, depending on the properties of the two antibody tests, the assumption of conditional independence of test results given disease status may not hold.

In the Norrbotten study, the non age-stratified prevalence was estimated at $\hat{f}^*_{1 \wedge 2} = 1.9\%$; other details of the testing protocols employed in this study are discussed in the Materials and methods section. We use the $\hat{f}^*_{1 \wedge 2}$ estimate to calculate the measured, error-corrected prevalence using Eq (39) and the underlying individual test sensitivities and specificities given in the Materials and methods section. We also calculate the corresponding 95% CIs by generating $10^6$ samples from beta distributions capturing the measured prevalence $\hat{f}^*_{1 \wedge 2}$. We present the measured prevalences $\hat{f}^*_{1 \wedge 2}$ and the corresponding measured, error-corrected prevalences $\hat{f}$ for various age groups in Table 3.

Eq (39) yields a non-stratified, error-corrected prevalence $\hat{f} = 2.5\%$ (1.1—5.0%), which is higher than the measured prevalence $\hat{f}^*_{1 \wedge 2} = 1.9\%$ (0.8—3.7%). Because the sensitivity of tests combined using an AND function is lower compared to the sensitivity of the underlying constituent tests, the measured prevalence associated with this aggregation function usually underestimates the true prevalence. Hence, the measured, error-corrected prevalence is substantially larger in this example than the measured one.

**An application in fatality and hospitalization monitoring.** Prevalence estimates commonly arise in infection fatality and hospitalization ratios, which are useful measures for monitoring outbreak severity. For a given jurisdiction at time $t$, the infection fatality ratio $\text{IFR}(a_k, t)$ of the population of age in the interval $[a_k, a_{k+1})$ is

$$\text{IFR}(a_k, t) = \frac{D(a_k, t)}{f(a_k, t)N(a_k)} , \tag{41}$$

where $f(a_k, t)$ and $D(a_k, t)$ respectively denote the age-stratified true proportion of infected individuals at time $t$ and the total number of infection-caused fatalities up to time $t$ measured from the start of an outbreak and within the age interval $[a_k, a_{k+1})$. In the above definition, we assume that the overall population $N(a_k)$ of age in the interval $[a_k, a_{k+1})$ is constant in the time horizon of interest. The denominator $f(a_k, t)N(a_k)$ in Eq (41) quantifies the total number of age-stratified infections at time $t$ since the start of an outbreak (*i.e.*, current and prior infections).

The number of infection-caused fatalities, $D(a_k, t)$, may be difficult to infer because of various confounding factors. These factors include variations in protocols for attributing the cause of death, the existence of co-morbidities [67], and delays in reporting. In jurisdictions where underreporting is prevalent, statistics on excess deaths may offer a more accurate assessment of the overall death toll [45, 48].

**Table 3. Measured and error-corrected prevalence in Norrbotten, Sweden (May 25—June 5, 2020)** [56]. The error correction method we employed takes into account the two tests used in the seroprevalence study from Norrbotten: (i) the Abbott SARS-CoV-2 IgG kit and (ii) the Euroimmun Anti-SARS-CoV-2 ELISA (IgG). These tests have been combined using an AND function. We calculated the measured, error-corrected prevalence through Eq (39) and their corresponding 95% CIs by generating $10^6$ samples from beta distributions capturing the measured prevalence as well as the underlying individual test sensitivities and specificities. Details of the study are listed in the Materials and methods section.

| age group | measured prevalence $\hat{f}^*_{1 \wedge 2}$ | error-corrected prevalence $\hat{f}$ |
|---|---|---|
| 20–29 years | 6.6% (1.8—15.9%) | 8.8% (2.4—21.6%) |
| 30–64 years | 0.7% (0.1—2.7%) | 0.9% (0.1—3.3%) |
| 65–80 years | 2.1% (0.3—7.3%) | 2.8% (0.4—9.5%) |

Analogous to the IFR, the infection hospitalization ratio $\text{IHR}(a_k, t)$ of the population of age in the interval $[a_k, a_{k+1})$ in a given jurisdiction is

$$\text{IHR}(a_k, t) = \frac{H(a_k, t)}{f(a_k, t)N(a_k)} \, , \tag{42}$$

where $H(a_k, t)$ is the corresponding total number of age-stratified infection-caused hospitalizations up to time $t$ measured from the start of an outbreak. Because of the time lag between infection and resolution, both the IFR and IHR may underestimate the true burden of an outbreak, especially in the early stages when the number of new cases increases rapidly [68]. In Table 4, we summarize the main variables used in outbreak severity measures.

The true proportion of infections $f(a_k, t)$ used in the denominators of both IFR and IHR is usually difficult to quantify for large populations. We can thus employ prevalence estimates $\hat{f}(a_k, t)$ as derived in Eq (38) that are usually based on serological testing of random samples of the entire population. Estimated proportions of infections $\hat{f}(a_k, t)$ that have been obtained using serological tests can be assumed to be close to the true proportions $f(a_k, t)$ if antibody waning is negligible and if the population sample is unbiased and representative of the whole population.

We denote the corresponding IFR and IHR estimates by

$$\widehat{\text{IFR}}(a_k, t) = \frac{D(a_k, t)}{\hat{f}(a_k, t)N(a_k)} \, , \tag{43}$$

and

$$\widehat{\text{IHR}}(a_k, t) = \frac{H(a_k, t)}{\hat{f}(a_k, t)N(a_k)} \, , \tag{44}$$

respectively.

In the Norrbotten, Sweden (May 25—June 5, 2020) seroprevalence study [56], we assume that the error-corrected seroprevalence estimate $\hat{f} = 2.5\%$ (1.1—5.0%) obtained for the 20 to 80 year old subpopulation is reflective of the prevalence in the entire population of 249,614 individuals. Using the total number of 59 fatalities and 242 hospitalizations documented throughout the entire study duration, along with Eqs (43) and (44), we obtain $\widehat{\text{IFR}} = 0.9\%$ (0.5—2.2%) and $\widehat{\text{IHR}} = 3.8\%$ (1.9—9.1%). These values are lower than the fatality ratio of 1.2% (0.6—3.0%) and hospitalization ratio of 5.1% (2.6—12.1%) obtained with the uncorrected, measured prevalence $\hat{f}_{1\wedge 2}^* = 1.9\%$ (0.8—3.7%).

**Table 4. Main variables used in outbreak severity measures.** Population, fatality, hospitalization, and prevalence statistics are often reported for $N_a$ age intervals $[a_{k-1}, a_k)$ ($k \in \{1, \ldots, N_a\}$) with $a_k = a_0 + \sum_{\ell=1}^{k} \Delta a_\ell$. Here, $a_0$ is the smallest age value in the data set and $\Delta a_\ell$ is the width of the $\ell$-th age window. We assume that the population size $N(a_k)$ is constant in the considered time window. The closed interval $[0, 1]$ contains 0, 1, and all numbers in between, and $\mathbb{N}$ denotes the set of non-negative integers.

| Symbol | Definition |
|---|---|
| $N(a_k) \in \mathbb{N}$ | population of age in the interval $[a_k, a_{k+1})$ in a given jurisdiction |
| $D(a_k, t) \in \mathbb{N}$ | total number of infection-caused fatalities of age in the interval $[a_k, a_{k+1})$ in a given jurisdiction at time $t$ (measured from the start of an outbreak) |
| $H(a_k, t) \in \mathbb{N}$ | total number of infection-caused hospitalizations of age in the interval $[a_k, a_{k+1})$ in a given jurisdiction at time $t$ (measured from the start of an outbreak) |
| $f(a_k, t){:}[0, 1]$ | true proportion of infected individuals of age in the interval $[a_k, a_{k+1})$ at time $t$ in a given jurisdiction |

## Discussion

Repeating and aggregating results from diagnostic and screening tests can significantly enhance overall test performance. Given ongoing advancements in technology and the need to effectively manage future infectious disease outbreaks, the methods presented in this work, as well as potential future extensions, can improve both testing protocols and estimates of infectious-disease surveillance measures such as prevalence, infection fatality ratio (IFR), and infection hospitalization ratio (IHR). While our primary focus has been on aggregating test results within the context of infectious-disease surveillance, similar concepts hold broad clinical applicability, such as in diabetes testing [4, 5], medical imaging [6–8], and cancer screening [10, 11]. The complex clinical conditions are usually probed by tests performing multiclass discrimination, requiring generalizations of the ROC surface and other reduction schemes [69].

Starting from the aggregation of the results of two tests, we derived expressions for the sensitivity and specificity of combined tests, assuming their conditional independence. To quantify dependence effects among tests, we formulated Boole–Fréchet inequalities for the sensitivity and specificity of several $n$-test Boolean functions. Additionally, we examined dependence effects using a dataset of test results from nine antibody assays [58]. Furthermore, we quantified the potential for saving tests when employing series testing compared to parallel testing, without compromising sensitivity and specificity. We then discussed the strong dependence of the positive predictive value (PPV) (*i.e.*, the ratio of true positives to positive calls) and negative predictive value (NPV) (*i.e.*, the ratio of true negatives to negative calls) on the employed aggregation mechanism. For example, AND aggregation yields relatively large PPVs and NPVs at low prevalence values, while OR aggregation does so for higher prevalences.

Expressions of sensitivity and specificity for aggregations of results from more than two tests can also be derived. Because these expressions become very lengthy, we developed an algorithm capable of identifying the best way of aggregating results from a given set of tests in terms of efficient sensitivity-specificity pairs (*i.e.*, sensitivity-specificity values that lie on an ROC frontier). We applied this algorithm to three commonly used SARS-CoV-2 tests and demonstrated how their individual sensitivities and specificities can be significantly improved when combined. Finally, we established a connection between combined tests and prevalence estimates in infectious-disease surveillance. Such estimates are pertinent for computing measures like the IFR and IHR.

Although our work addresses various factors related to aggregating test results, there are additional aspects that we have not considered. For instance, certain tests may entail higher costs or varying levels of complications for patients (see, *e.g.*, chapter 7.4 in [38]). Other refinements may incorporate test-avoidance, or increasing levels of test-fatigue when multiple tests are to be administered. Incorporating these effects requires formulating appropriate objective functions and adjusting our optimization approach. In the context of an ROC curve, an objective function that quantifies the utility gain associated with treating a sick individual and the utility loss associated with treating a healthy individual enables the identification of the optimum aggregation approach [70]. Another interesting direction for future research is to generalize Boole–Fréche-type inequalities [63] to encompass a larger set of aggregation functions. Moreover, although we have incorporated sensitivity and specificity data for numerous tests in our analysis, it would be worthwhile to further validate the results of our model through experimental data on aggregated test results.

In addition to the described applications, our work can help inspire aggregation methods in social choice theory and decision-making under uncertainty, where the objective is to effectively combine individual opinions [71–75]. For instance, it can inform decision-making processes in organizations where decision makers also possess sensitivities and specificities with

respect to a given decision task. Furthermore, our work is closely connected to contributions on fault-tolerant computing by von Neumann [76], Moore, and Shannon [77–79], who studied how reliable (Boolean) computing elements can be constructed from unreliable components.

## Materials and methods

### Dependence factors

To quantify dependencies between tests, we analyze results from nine lateral flow immunoassay (LFIA) devices using plasma samples from individuals with confirmed COVID-19 based on PCR results, as well as pre-pandemic negative control samples collected in the UK before December 2019. The data are publicly available in [58]. Due to the limited availability of LFIA devices, not all tests could be performed on every sample. The median sensitivity of the LFIA devices ranges from 0.55 to 0.70, while the specificity ranges from 0.95 to 1.00.

Given the large number of possible combinations of subsets of these nine LFIA tests, we focus on examining two dependence factors associated with aggregating results from two tests. The outlined method can be applied to other combinations as well.

The joint probability mass function $\Pr(Y_i = 1, Y_j = 1 \mid X = 1)$, for distinct tests $i$ and $j$ (where $i, j \in \{1, \ldots, 9\}$ and $i \neq j$), is

$$
\begin{aligned}
\Pr(Y_i = 1, Y_j = 1 \mid X = 1) &= \Pr(Y_i = 1 \mid X = 1)\Pr(Y_j = 1 \mid Y_i = 1, X = 1) \\
&= \Pr(Y_j = 1 \mid X = 1)\Pr(Y_i = 1 \mid Y_j = 1, X = 1) .
\end{aligned}
\tag{45}
$$

To account for a linear dependence between $\Pr(Y_i = 1 \mid Y_j = 1, X = 1)$ and $\Pr(Y_i = 1 \mid X = 1)$, we set $\Pr(Y_i = 1 \mid Y_j = 1, X = 1) = \lambda_{11|1}^{(ij)}\Pr(Y_i = 1 \mid X = 1)$. Recall that Eq (45) describes the TPR of a two-test AND protocol.

Next, we calculate $\Pr(Y_i = 1, Y_j = 1 \mid X = 1)$ for all $\binom{9}{2} = 36$ combinations of tests, and then we compute the corresponding dependence factors

$$
\lambda_{11|1}^{(ij)} = \frac{\Pr(Y_i = 1, Y_j = 1 \mid X = 1)}{\Pr(Y_i = 1 \mid X = 1)\Pr(Y_j = 1 \mid X = 1)} ,
\tag{46}
$$

which satisfy $\lambda_{11|1}^{(ij)} = \lambda_{11|1}^{(ji)}$. Likewise, we calculate the dependence factors

$$
\lambda_{00|0}^{(ij)} = \frac{\Pr(Y_i = 0, Y_j = 0 \mid X = 0)}{\Pr(Y_i = 0 \mid X = 0)\Pr(Y_j = 0 \mid X = 0)} .
\tag{47}
$$

Recall that $\Pr(Y_i = 0, Y_j = 0 \mid X = 0)$ describes the TNR of a two-test OR protocol.

In Fig 7, we show the distributions of $\lambda_{11|1}^{(ij)}$ and $\lambda_{00|0}^{(ij)}$ derived from the empirical data in [58]. The mean values, $\bar{\lambda}_{11|1}^{(ij)}$ and $\bar{\lambda}_{00|0}^{(ij)}$, are 1.42 and 0.99, respectively. This suggests that, on average, $\Pr(Y_i = 1 \mid Y_j = 1, X = 1)$ is about 40% larger than $\Pr(Y_i = 1 \mid X = 1)$, while $\Pr(Y_i = 0 \mid Y_j = 0, X = 1)$ is roughly equal to $\Pr(Y_i = 0 \mid X = 0)$.

### Boole–Fréchet inequalities

If individual tests exhibit conditional dependencies, the Boole–Fréchet inequalities [60–63] can be used to establish lower and upper bounds for the sensitivities and specificities of aggregated test results.

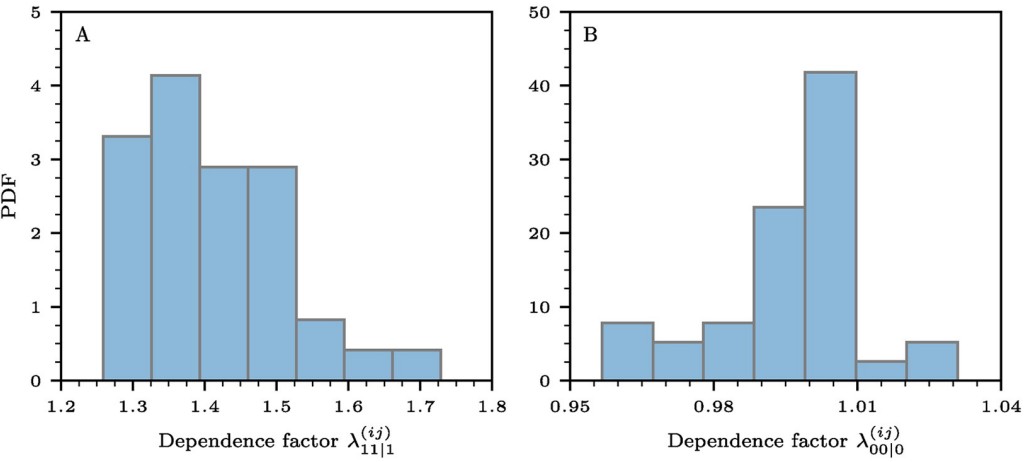

**Fig 7. Probability density functions (PDFs) of dependence factors (A) $\lambda_{11|1}^{(ij)}$ (see Eq (46)) and (B) $\lambda_{00|0}^{(ij)}$ (see Eq (47)).**

For example, for an AND aggregation function, we have

$$\max\left(0, \sum_{i=1}^{n} \text{TPR}_i - (n-1)\right) \leq \text{TPR}_{1\wedge\cdots\wedge n} \leq \min_i(\text{TPR}_i) \tag{48}$$

and

$$\max_i(\text{TNR}_i) \leq \text{TNR}_{1\wedge\cdots\wedge n} \leq \min\left(1, \sum_{i=1}^{n} \text{TNR}_i\right). \tag{49}$$

Similarly, for an OR aggregation function, we have

$$\max_i(\text{TPR}_i) \leq \text{TPR}_{1\vee\cdots\vee n} \leq \min\left(1, \sum_{i=1}^{n} \text{TPR}_i\right) \tag{50}$$

and

$$\max\left(0, \sum_{i=1}^{n} \text{TNR}_i - (n-1)\right) \leq \text{TNR}_{1\vee\cdots\vee n} \leq \min_i(\text{TNR}_i). \tag{51}$$

These inequalities do not assume any specific dependence structure between the individual tests and provide the best possible bounds when only the sensitivities and specificities of the individual tests are known [63].

For an odd number of tests, we also consider the majority function. The majority function interpolates between the extremes of requiring all tests to be positive (AND) and requiring at least one positive result (OR). We thus have $\text{TPR}_{1\wedge\ldots\wedge n} \leq \text{TPR}_{M(1,\ldots,n)} \leq \text{TPR}_{1\vee\ldots\vee n}$ and $\text{TNR}_{1\vee\ldots\vee n} \leq \text{TNR}_{M(1,\ldots,n)} \leq \text{TNR}_{1\wedge\ldots\wedge n}$. Hence, the majority function satisfies

$$\max\left(0, \sum_{i=1}^{n} \text{TPR}_i - (n-1)\right) \leq \text{TPR}_{M(1,\cdots,n)} \leq \min\left(1, \sum_{i=1}^{n} \text{TPR}_i\right) \tag{52}$$

and

$$\max\left(0, \sum_{i=1}^{n}\mathrm{TNR}_i - (n-1)\right) \leq \mathrm{TNR}_{\mathrm{M}(1,\cdots,n)} \leq \min\left(1, \sum_{i=1}^{n}\mathrm{TNR}_i\right). \tag{53}$$

### Beta distribution sampler

To calculate CIs associated with combined tests and related quantities that depend on multiple factors such as sensitivity, specificity, and prevalence, we employ a Monte Carlo sampling technique. In this work, we consider samples drawn from a beta distribution

$$\mathbb{P}(x; \alpha, \beta) = \frac{\Gamma(\alpha)\Gamma(\beta)}{\Gamma(\alpha + \beta)} x^{\alpha-1}(1-x)^{\beta-1}, \tag{54}$$

where $x \in [0, 1]$, $\alpha, \beta$ are shape parameters, and $\Gamma(\cdot)$ denotes the gamma function. Sensitivities, specificities, and prevalences are quantities with a support of $[0, 1]$, so beta distributions are plausible approximations of their underlying distributions.

We determine shape parameters such that the corresponding distributions capture the median and 95% CIs of the underlying quantities. To do so, we minimize the sum of squared differences between the cumulative distribution at the 2.5%, 50%, and 97.5% quantiles, and the corresponding empirical median and 95% CI values. We carry out this optimization process by employing the `fmin` function implemented in `scipy.optimize` in `Python`. Further implementation details are available at [80].

### Hospitalization, fatality, and serology data

In the main text, we use data from a seroprevalence study conducted in Norrbotten, Sweden, during weeks 22 and 23 of 2020 (May 25 to June 5) [56]. We considered the hospitalization, fatality, and seroprevalence data provided in this study to illustrate how errors associated with combined tests can be addressed. The study encompassed a population of 182,828 adults aged 20 to 80 years. The age distribution within this population was as follows: 16.2% were aged 20 to 29 years, 57.8% were aged 30 to 64 years, and 25.9% were aged 65 to 80 years. From this population, 500 individuals were randomly selected and contacted, out of which 425 participated in the study. A total of 242 individuals with confirmed infection had been hospitalized since the beginning of the outbreak, and 59 people with confirmed infection had passed away.

The study revealed a population-wide measured prevalence $\hat{f}^*_{1\wedge2}$ of 1.9% (0.8—3.7%). Seroprevalence was assessed using two different assays: (i) the Abbott SARS-CoV-2 IgG kit and (ii) the Euroimmun Anti-SARS-CoV-2 ELISA (IgG). The former has a sensitivity and specificity of 83.1% (75.4—100%) and 100%, respectively [81]. The sensitivity and specificity of the latter are 91.1% (80.7—96.1%) and 100% (96.5—100%), respectively [82].

Every individual who tested positive in Abbott's assay underwent confirmation using Euroimmun's Anti-SARS-CoV-2 ELISA (IgG). This process represents an AND aggregation.

### Acknowledgments

The authors thank Stefan Felder and Nana Owusu-Boaitey for helpful comments.

### Author Contributions

**Conceptualization:** Lucas Böttcher.

**Data curation:** Lucas Böttcher.

**Formal analysis:** Lucas Böttcher, Maria R. D'Orsogna, Tom Chou.

**Funding acquisition:** Lucas Böttcher, Maria R. D'Orsogna.

**Methodology:** Lucas Böttcher, Maria R. D'Orsogna, Tom Chou.

**Software:** Lucas Böttcher.

**Visualization:** Lucas Böttcher.

**Writing – original draft:** Lucas Böttcher.

**Writing – review & editing:** Lucas Böttcher, Maria R. D'Orsogna, Tom Chou.

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
