## [Decision Letter · Decision Letter 0]

4 Oct 2024

Dear Professor Böttcher,

Thank you very much for submitting your manuscript "Aggregating multiple test results to improve medical decision-making" for consideration at PLOS Computational Biology.

As with all papers reviewed by the journal, your manuscript was reviewed by members of the editorial board and by several independent reviewers. In light of the reviews (below this email), we would like to invite the resubmission of a significantly-revised version that takes into account the reviewers' comments.

We cannot make any decision about publication until we have seen the revised manuscript and your response to the reviewers' comments. Your revised manuscript is also likely to be sent to reviewers for further evaluation.

Sincerely,

Mark Alber, Ph.D.

Section Editor

PLOS Computational Biology

Mark Alber

Section Editor

PLOS Computational Biology

Reviewer's Responses to Questions

**Comments to the Authors:**

Reviewer #1: Please see attached review.

Reviewer #2: The paper discusses a statistical model designed to enhance medical decision-making by addressing uncertainties that arise from diagnostic errors—specifically, type I (false positive) and type II (false negative) errors. The model aims to improve decision accuracy by repeating and aggregating the results of diagnostic and screening tests. This methodology is applicable in both clinical settings (such as medical imaging) and public health scenarios, exemplified by the need for efficient testing during the SARS-CoV-2 pandemic.

The paper is well-written. I checked the mathematics and found it solid. However, there are some concerns as follows.

Practical implementation of the model in real-world settings might be complex and require substantial computational resources. The authors may want to explain this with empirical data.

The effectiveness of the model may depend on the availability and quality of observational data. This paper is largely theoretical without extensive experiments with real data. More experiments to demonstrate its utility is required before it can be published.

Similarly, while the model is designed to be adaptable, its performance across different diseases and testing scenarios would need thorough validation.

**Have the authors made all data and (if applicable) computational code underlying the findings in their manuscript fully available?**

Reviewer #1: Yes

Reviewer #2: Yes

PLOS authors have the option to publish the peer review history of their article (what does this mean?). If published, this will include your full peer review and any attached files.

Reviewer #1: No

Reviewer #2: No
---

## [Decision Letter · Decision Letter 1]

25 Dec 2024

Dear Professor Böttcher,

We are pleased to inform you that your manuscript 'Aggregating multiple test results to improve medical decision-making' has been provisionally accepted for publication in PLOS Computational Biology.

Best regards,

Mark Alber, Ph.D.

Section Editor

PLOS Computational Biology

Mark Alber

Section Editor

PLOS Computational Biology

Reviewer's Responses to Questions

**Comments to the Authors:**

Reviewer #1: I thank the authors for taking the time to consider my comments carefully. I especially appreciate the new content on the Boole-Frechet inequalities.

Reviewer #2: The authors have addressed all issues that I raised in the first round of review. I would happily recommend acceptance of the paper.

**Have the authors made all data and (if applicable) computational code underlying the findings in their manuscript fully available?**

Reviewer #1: Yes

Reviewer #2: Yes

PLOS authors have the option to publish the peer review history of their article (what does this mean?). If published, this will include your full peer review and any attached files.

Reviewer #1: No

Reviewer #2: **Yes: **Qingpeng Zhang

---

## [Editor Report · Acceptance letter]

30 Dec 2024

PCOMPBIOL-D-24-00885R1 

Aggregating multiple test results to improve medical decision-making

Dear Dr Böttcher,

I am pleased to inform you that your manuscript has been formally accepted for publication in PLOS Computational Biology. Your manuscript is now with our production department and you will be notified of the publication date in due course.

With kind regards,

Zsofia Freund
